# The Health Status of the US Veterans: A Longitudinal Analysis of Surveillance Data Prior to and during the COVID-19 Pandemic

**DOI:** 10.3390/healthcare11142049

**Published:** 2023-07-17

**Authors:** Jose A. Betancourt, Diane M. Dolezel, Ramalingam Shanmugam, Gerardo J. Pacheco, Paula Stigler Granados, Lawrence V. Fulton

**Affiliations:** 1School of Health Administration, Texas State University, San Marcos, TX 78666, USA; shanmugam@txstate.edu (R.S.); gjp46@txstate.edu (G.J.P.); 2Department of Health Information Management, Texas State University, Round Rock, TX 78665, USA; dd30@txstate.edu; 3School of Public Health, San Diego State University, San Diego, CA 92182, USA; pstiglergranados@sdsu.edu; 4Woods College of Advancing Studies, Boston College, Chestnut Hill, MA 02467, USA; lawrence.fulton@bc.edu

**Keywords:** US veteran health, comorbidities, risk-factors, military readiness, COVID-19

## Abstract

Chronic diseases affect a disproportionate number of United States (US) veterans, causing significant long-term health issues and affecting entitlement spending. This longitudinal study examined the health status of US veterans as compared to non-veterans pre- and post-COVID-19, utilizing the annual Center for Disease Control and Prevention (CDC) behavioral risk factor surveillance system (BRFSS) survey data. Age-adjusted descriptive point estimates were generated independently for 2003 through 2021, while complex weighted panel data were generated from 2011 and onward. General linear modeling revealed that the average US veteran reports a higher prevalence of disease conditions except for mental health disorders when compared to a non-veteran. These findings were consistent with both pre- and post-COVID-19; however, both groups reported a higher prevalence of mental health issues during the pandemic years. The findings suggest that there have been no improvements in reducing veteran comorbidities to non-veteran levels and that COVID-19 adversely affected the mental health of both populations.

## 1. Introduction

United States (US) veterans have worse health outcomes than non-veterans, even after adjusting for age and other factors [1]. Previous studies have identified that obesity, diabetes, heart disease, stroke, skin cancer, other cancers, and chronic obstructive pulmonary disease (COPD) are statistically greater among veterans than non-veterans, even after age adjustment [1]. Potential reasons why veterans suffer from chronic diseases more than the general population include physical and psychological stress from combat duty and deployments, physical or mental trauma, and the necessity to maintain the required levels of physical fitness for active duty [2,3]. For example, it is well established that stress can lead to loss of sleep, unhealthy eating, and poor exercise habits, which are associated with many chronic illnesses.

A 2017 study found that arthritis was more prevalent in the veteran community [4]. Additionally, chronic kidney disease (CKD) is a serious condition associated with kidney failure, cardiovascular disease, and an increased probability of mortality. Veterans have a higher prevalence of CKD; approximately one in six veterans have CKD as compared to one in seven Americans with CKD [5]. CKD risk factors include diabetes, high blood pressure, age over 60 years, obesity, heart disease, and race (i.e., African American, Hispanic, Native American, or Asian) [5]. Similarly, a retrospective study of 2006–2015 US Military System data revealed higher rates of CKD for older veterans, blacks, and senior enlisted ranks [6].

Mental health issues are also prevalent among veterans. Approximately 43% of veterans in a large study experienced mental health issues, including post-traumatic stress disorder (PTSD), depressive disorders, or substance abuse [7]. Depression is a serious diagnosis that has been associated with higher risks of mortality and morbidity [8]. Prior research on data from the 2005–2016 national health and nutrition examination surveys (NHANES) determined that the pooled prevalence of depression in veterans was higher in comparison to the prevalence of depression for non-veteran adults with at least one major depressive disorder reported in the national survey on drug use and health [8]. Disparities in the prevalence of depression were identified in the NHANES study; females, White veterans, and veterans with low income levels or lower education levels had higher rates of depression [8]. In general, veterans have 1.5 times higher suicide rates than the general US population [9]. Additionally, veterans with mental health issues are especially at risk for suicide when compared to non-veterans [8,10].

The chronic diseases and mental health conditions discussed in this paper not only affect a disproportionate number of veterans but are also known to cause significant long-term health issues and negatively impact the US economy. According to the Centers for Disease Control and Prevention (CDC), expenses for individuals with chronic diseases and mental health conditions account for 90% of the annual US healthcare expenses [11]. In particular, heart disease and stroke cost the healthcare system USD 216 billion per year and account for the majority (33%) of annual disease-specific mortalities [11].

### COVID-19 Impacts on Health

The impact of the COVID-19 global pandemic on the health and well-being of veteran and non-veteran populations in the US is still being studied. This phenomenon disrupted healthcare and access to services, caused social isolation, and created many other disturbances that impacted the health of different populations. These issues disproportionately impact those who are of racial/ethnic minorities, have lower socioeconomic status, have less education, and are veterans [12]. Emerging research on veterans has shown that COVID-19 inflicts post-traumatic stress disorder (PTSD) among some veterans [13]. Consequently, some veterans resort to using alcohol and cannabis, according to a six-month follow-up survey conducted during and after the COVID-19 pandemic [14]. Linjawi et al. (2023) pointed out that the pandemic adversely affected cancer patients because of the shortage of medical supplies, beds, and healthcare workers in hospitals [15].

Specifically, Resendes et al. (2023) explained the impact of COVID-19 on the frail, which is an age-related syndrome for hospitalization [16]. Weaver et al. (2022) reported that one out of every six veterans who were diagnosed with COVID-19 was re-admitted to the hospital within 90 days [17]. The longer the hospital stay, the greater the likelihood of re-admission, especially when the veteran had multiple comorbidities, smoked, or lived in an urban area. McGuire et al. (2023) found that veterans’ social isolation since COVID-19 became significant and resulted in worse mental health outcomes [18]. Gujral et al. (2023) pursued the impact of COVID-19 on veterans’ suicide risk within the healthcare system [19]. Akhtar et al. (2023) summarized that COVID-19 increased the risk of cardiovascular illness [20].

This research extends a previous study [1], which provided descriptive age-adjusted point estimates of the health status of US veterans versus non-veterans for the years 2003-2019. This study adds to the data for 2020 and 2021 to highlight any population differences that may have occurred as a result of the global pandemic. As in the previous study, specific dependent variables of interest included the reported prevalence of obesity, heart disease, stroke, skin cancer, other cancers, COPD, arthritis, mental health, kidney disease, and diabetes. The CDC advises against combining data before 2011 due to changes in its behavioral risk factor surveillance system (BRFSS) survey [21], so the inferential analysis was restricted to the combined surveys from 2011 through 2021. It was hypothesized that changes in observed prevalence over time would be different for veterans based on previous work [1]. As a secondary aim, we explored if there were changes in the prevalence of the selected outcomes of interest pre- and post-COVID-19 between veterans and non-veterans.

## 2. Materials and Methods

BRFSS data sets from 2003 through 2021 provided by the CDC were analyzed. BRFSS is an annual state- and territory-based health survey that covers adults over the age of 18 in the United States, three US territories (Puerto Rico, the US Virgin Islands, and Guam), and the District of Columbia [22]. The survey relies on household and cellular telephone surveys of non-institutionalized adults. Approximately 400,000 surveys per year are conducted, and the survey is publicly available through the CDC [21,23,24]. These data sets provide information about chronic diseases in the US and use complex sampling methods requiring equally complex weighting schemes [21,23,24].

Data from 2003 through 2021 (the latest available year) were used to evaluate comparatively US Veteran versus non-veteran disease prevalence. Of specific interest was the prevalence of obesity, heart disease, stroke, skin cancer, other cancers, COPD, arthritis, mental health, kidney disease, and diabetes. Further, we investigated whether there were prevalence changes for either or both groups in the pre- and post-COVID-19 periods defined as before 2020 versus 2020 through 2021, the latest year of available data. Quasibinomial general linear models (GLM) investigated the effects of demographics, socioeconomic factors, geographic region, year, veteran status, COVID-19 period, and the interaction between COVID-19 and Veteran status.

Using the BRFSS sampling weights, age-adjusted disease prevalence was estimated for each year individually from 2003 through 2021 to provide descriptive insight. The years 2011 through 2021 were then estimated together as a panel series. Previous years (2003 through 2010) were not used in the panel series due to BRFSS sampling changes [21]. The 2003 through 2010 data are not directly comparable to more recent data (e.g., 2011 through 2021) because of changes in the weighting methodology and the addition of cell phone sampling. The pre-2011 surveys do provide the best available point estimates for those years and are thus used for descriptive analysis (uncombined).

### 2.1. Sample

Table 1 provides the unweighted sample and the weighted population estimates for veterans and non-veterans from 2011 through 2021. The unweighted data are the BRFSS data that were collected with the phone surveys. When BRFSS surveys are conducted, some areas may not have good survey coverage, have a low response rate, or exhibit variations in response rates. BRFSS data-weighting adjusts for these conditions by making the total number of cases (i.e., survey responses) equal to the population estimates for each geographic region (state, territory, or district), which in turn provides a more accurate representation of the population [22,24].

The total veteran population estimate for all study years was 465,008,694, and the total non-veteran population estimate was 3,610,450,803. For all study years, an average of 11.41% of the population identified as veterans. The average percentage of veterans represented in each annual estimated population survey declined across the years, from 13.62% in 2003 to 9.73% in 2021.

### 2.2. Study Variables

Table 2 displays dependent variables, independent variables, and demographic, socioeconomic, and graphic controls in the study. It also shows the coding and description for each variable.

#### 2.2.1. Dependent Variables

For the dependent variables aside from ‘mental health’, respondents were provided the opportunity to respond ‘yes’, ‘no’, and ‘do not know/not sure’, and some observations were missing or not asked. The proportion missing for each of the dependent variables except for ‘overweight/obese’ status was less than 1%. Approximately 10% of overweight/obese responses were missing. In all cases, the statistical mode was imputed for these variables. For the ‘mental health’ variable, respondents were asked to self-report the number of days in the last 30 days that their mental health, which includes stress, depression, and problems with emotion, was not good. We re-coded the mental health variable to be dichotomous, with ‘1’ indicating that a respondent reported any number of days that their mental health was not good and ‘0’ otherwise.

#### 2.2.2. Independent Variables

The first independent variable, veteran status, was coded identically for all years except 2009. Collapsing this variable to ‘yes’ and ‘other than yes’ provided the consistency needed for analysis. There were less than 1% of these responses that were blank, whereas the blank responses were imputed with the modal response of a non-veteran.

The second independent variable, COVID-19, was a linear spline variable. For years prior to 2019, the value of the spline was zero. For 2020 and 2021, the values were 1 and 2, respectively. The use of splines allows for a separate slope once an event has occurred. For example, assume that the result of a GLM is *Y = 10 + 5 × COVID*. Before 2020, the estimate for this model would be 10. For 2020 and 2021, the estimates would be 15 and 20, respectively (a linear change). Thus, the spline allows for a secondary slope associated with COVID-19.

Finally, an interaction term between COVID-19 and veteran status was included. This interaction term helps differentiate effects based on veteran versus non-veteran status pre- and post-COVID-19.

#### 2.2.3. Control Variables

Demographic Variables

Demographic control variables included categorical age and race variables as well as gender (1 = male, 0 = otherwise) and marital status (1 = married, 0 = otherwise). Race was recoded into two variables: race (1 = Caucasian, 0 = Otherwise) and ethnicity (1 = Hispanic, 0 = Otherwise). Age was already an imputed variable with coded categories: 1 = 18–24, 2 = 25–34, 3 = 35–44, 4 = 45–54, 5 = 55–64, and 6 = 65+.

Socioeconomic Variables

Socioeconomic controls included income, education, and employment. All three were evaluated as dichotomous variables. Income was coded as 1 = USD 75,000 or more, 0 = otherwise. For education, the coding was 1 = college graduate, 0 = otherwise. Employment was recategorized as 1 = employed for wages (modal response), 0 = otherwise.

Geographic Variable

One geographic control variable, ‘Census Division’, was created for this study by collapsing states into divisions. Specifically, we collapsed the BRFSS ‘State’ variable, based on federal information processing codes, into census bureau divisions as follows: 1 = New England, 2 = East North Central, 3 = East South Central, 4 = Middle Atlantic, 5 = Mountain, 6 = Pacific, 7 = South Atlantic, 8 = West North Central, 9 = West South Central, 10 = Territories.

Time Variable

A variable (Year) provided a temporal measure for analyzing changes over time. We postulated that linear effects might exist for the period analyzed (2011–2021).

### 2.3. Model and Methods

#### 2.3.1. Descriptive Models

For each dependent variable and each year, we built age-adjusted models of percentages reported by health condition by year and by veteran status, to evaluate differences between veteran and non-veteran reported health issues.

#### 2.3.2. General Linear Models

Our study team also built a quasi-binomial general linear model using the 2011–2021 data. The quasibinomial allows for non-integer binomial responses and adds a separate measure for dispersion not totally measured by the binomial [25]. All analyses were conducted in R Statistical software version 4.3.1 using *epitools* for age-adjustments and the *survey* package for applying complex weighting [26,27]. For the unweighted data, logistic regression would have been appropriate. However, when the survey package weights are applied to the BRFSS data for population estimation, the dichotomous dependent variables could have fractional values. Thus, general linear models with a quasi-binomial link function from the R survey package were used for the weighted survey analysis because they allow fractional values of integers.

## 3. Results

### 3.1. Descriptive Statistics

Table 3 presents the age-adjusted comparisons of all the dependent variables for veterans and non-veterans by year, rolled up for years 2011 through 2021. The years 2003–2010 are displayed, but as mentioned, they cannot be combined due to changes in the BRFSS data weighting and data collection by the CDC. At the far right of Table 3, referencing the combined years in the Y11-21 column, we see that for this period veterans had higher rates of self-reported obesity (71% vs. 60%), diabetes (19% vs. 16%), heart disease (12% vs. 7%), stroke (6% vs. 5%), skin cancer (17% vs. 12%), other cancers (13% vs. 12%), COPD (10% vs. 9%), and kidney disease (5% vs. 4%).

Both groups have the same age-adjusted percentages for arthritis (37%) for the combined years in Y11-2, but the mental health (depression) rates were lower for veterans (31% vs. 38%). Indeed, the mental depression rates were lower for veterans for each year in this analysis. A review of the age-adjusted table values indicates that the trends for the variables across the years 2003 to 2021 show little variance.

Of interest is that mental health issues (depression) for both groups slowly increased between 2019 and 2021, the years of COVID-19. (COVID-19 appeared in China in November 2019 and in the US in January 2020, meaning that two survey months were included in the BRFSS responses for that year). Before 2019, the percentage of veterans and non-veterans reporting mental health issues was 30.06% and 36.50%, respectively. From 2019 onward, the percentages increased to 34.25% and 40.75%, respectively.

### 3.2. General Linear Model, 2011 through 2021

Table 4 displays the quasibinomial General Linear Models (GLM) for each of the dependent variables with indicator variables for the years 2011–2021. For the GLM, the dependent variables were the disease processes and mental health status; there was one GLM generated for each of these comorbidities. All possible models with the inclusion and exclusion of independent variables were tried. It so happened that all the models fit well and, hence, are acceptable. Additionally, variance inflation factors (VIFs) for all variables and all models were below 2.0. Our study team built complete models given the size of the sample. The results illustrate that most variables are statistically significant. Those that are not provide compelling information given that traditional statistical significance is linked to sample size. By using structured model building, this reinforces the stability of the odds-ratio directionality and magnitude; however, this is omitted.

GLM results show the odds ratios for the control variables (demographic, socioeconomic, and geographic), veteran status, and year the data were collected. The control variables increase the internal validity by limiting the effect of extraneous variables associated with the comorbidities and with geographic location (e.g., culture).

#### 3.2.1. Veteran Status

Odds ratios (OR) greater than 1.0 may indicate the disease/event/illness is more likely to occur for any respondents in that category, and OR less than 1.0 may indicate it is less likely to occur, depending on statistical significance. Examining the variable ‘Veteran’ in Table 4, we see that veterans have higher odds of having every health condition in the table when compared to non-veterans except for mental health disorders, which are not significantly different from those of non-veterans. Specifically, the odds ratio for mental health is 0.992. These results agree with the age-adjusted percentages by health condition by year (Table 3), which indicate the self-reported percentages for depression were lower for veterans when compared to non-veterans.

#### 3.2.2. COVID Spline

Further, the pre- and post-COVID-19 splines suggested mixed effects across the dependent variables. Specifically, obesity/overweight and mental health issues had statistically significant ORs that were greater than 1.0 (1.008 and 1.069, respectively). Stroke, skin cancer, cancer, COPD, arthritis, kidney disease, and diabetes had statistically significant ORs less than 1.0. The spline was not significant for heart disease.

#### 3.2.3. Interaction between Veteran Status and Pre/During COVID-19

When considering the interaction between the COVID-19 spline and veterans, veterans were less likely than non-veterans to report obesity/overweight during COVID-19. Veterans were, however, more likely during COVID-19 to report skin cancer, cancer, COPD, arthritis, and kidney disease.

#### 3.2.4. Control Variables

Table 4 also displays results for the socioeconomic, demographic, geographic, and temporal control variables. Referencing Appendix A, the odds for all diseases were homogeneous during the initial age duration of 25 through 34 but drifted significantly to more heterogeneity at a later age. A higher number for the odds ratio indicates a higher risk for veterans.

Results indicate that the odds of obesity, heart disease, stroke, skin cancer, other cancers, COPD, arthritis, kidney disease, or diabetes generally increase with age (see Appendix A). Conversely, the risk of mental health disorders declines. Caucasians had higher risks of having heart disease, skin cancer, other cancers, COPD, and arthritis and lower risks of all other diseases. Hispanics had higher odds for overweight/obesity, skin cancer, and diabetes. Males were more likely to report overweight/obese status, heart disease, stroke, skin cancer, and diabetes. Married individuals were more likely to report being overweight/obese and report skin cancer.

Those with an income greater than USD 75,000 were more likely to report being obese/overweight and having skin cancer. College graduates were more likely to report skin cancer or other cancers. Wage employees were more likely to report being obese/overweight.

Large regional variations existed for each dependent variable (Region 1 = referent group). Region 2 had the highest odds for overweight/obesity, stroke, cancer, COPD, arthritis, kidney disease (tied with Region 10), and diabetes. Region 8 had the highest odds of heart disease, while Region 7 had the highest odds of skin cancer. For mental health issues, Region 6 had the highest odds ratio.

Odds ratios for all years were greater than 1.0 for stroke, skin cancer, COPD, mental health disorders, kidney disease, and diabetes for the veterans. For obesity/overweight status and arthritis, odds ratios were lower than 1.0 for the veterans.

## 4. Discussion

This longitudinal study aimed to assess age-adjusted percentages of selected health conditions by year between veterans and non-veterans and explore factors that may be associated with these outcomes among these groups. The results show that overall, veterans have higher odds of all investigated morbidities other than mental health diseases. Conversely, the odds of diseases during COVID-19 were lower for all morbidities, with the exception of mental health. The reason for these lower odds is unknown but may be attributable to delayed health care and the discovery of these morbidities. Mental health issues associated with COVID-19 are well known, and the higher odds ratio is expected. These results agree with previous studies that found veterans have a lower health status than non-veterans [1,4,5,6,7,8].

The interaction between Veteran status and the COVID-19 spline produced some interesting results. Obesity/overweight, skin cancer, COPD, arthritis, and kidney disease odds ratios, while still higher for veterans, changed at a lower rate than those of non-veterans. In agreement, Riveria et al. (2017) determined arthritis was more prevalent among veterans and was the primary reason for medical discharge among combat veterans [4]. The higher levels of arthritis may result from the physical stress of combat duty and the need to maintain a high level of physical fitness for active duty [2]. Higher obesity/overweight levels among veterans are concerning because they are associated with a higher risk of cancer (e.g., colon, esophageal, and thyroid), cardiovascular diseases, diabetes, osteoarthritis, osteoporosis, and morbidity [3,28,29,30,31]. Regarding COPD, it has been suggested that the effects of open-air trash burn pits, common in foreign duty stations, and service-related inhalants (e.g., combat dust, sandstorms, and occupational gases) could account for the higher odds of pulmonary diseases among veterans [32]. Moreover, higher rates of lung cancer among veterans may be related to their higher COPD rates because COPD is a significant risk factor for lung cancer [31]. Overall, the stressors associated with military service and combat duty, both physical and mental, contribute to the overall higher rates of the investigated diseases among veterans compared to non-veterans.

The results also illustrate that the odds of reporting the investigated diseases generally increase with age (until age 65+ where obesity/overweight status and COPD decline), except for mental health odds (which decrease by age group). These results agree with several other studies on the positive relationship between age and the reporting of chronic diseases. Niakouei et al. (2020) found that higher age groups were more likely to report heart disease [33]. Oliver et al. (2022) reported higher rates of CKD among older veterans [6]. A related study using BRFSS data determined that the odds of obesity, heart disease, skin cancer, other cancers, COPD, arthritis, kidney disease, and diabetes increased with age among veterans [1].

Race, gender, income, educational attainment, and unemployment were predictors of several chronic diseases. Caucasians report higher odds versus other groups of all investigated morbidities other than obesity/overweight status, stroke, kidney disease, and diabetes. Hispanics reported lower odds of heart disease, stroke, cancer, COPD, arthritis, and mental health disorders, which is consistent with results from other studies on veterans [1,34].

Males as compared to females reported higher odds ratios for cancer, arthritis, mental health disorders, and kidney disease. With the exception of overweight/obesity status, those making more than USD 75,000 per annum had lower odds of morbidity. These results agree with Oliver et al. (2022), who reported that females had a lower risk of CKD [6]. Similarly, Walker et al. (2019) determined that gender and income were both positively associated with heart disease [34]. Indeed, individuals with lower incomes may be challenged to maintain health insurance and may not be able to afford the medicines, doctor’s visits, or routine lab tests that can assist with chronic disease detection and management.

Wage employees versus self-employed, unemployed, or other statuses had lower odds of morbidity. Except for cancer, college graduates also had lower odds of morbidity. Those who reported being married were associated with lower odds in every category except for obesity/overweight status and skin cancer. Niakouei et al. (2020) established that unemployment was associated with a higher risk of cardiovascular events, while higher incomes and higher educational attainment were related to a lower risk of cardiovascular diseases (which include heart disease) [33]. This is intuitive because employed individuals are more likely to have health insurance and income for health maintenance activities such as going to the gym or eating a balanced diet. Conversely, unemployment is stressful and may lead to poor diet choices and exercise habits, smoking, or obesity, which increase the risks of many chronic diseases. In other studies, lower educational attainment was correlated with low health literacy, which contributes to poor health outcomes, missed appointments, medication non-compliance, and difficulty managing chronic diseases [35,36,37,38,39].

Geospatial effects have been explored for states or regions by other researchers [25,28,40]. Razzaghi et al. (2020) considered the county-level prevalence of medical conditions associated with increased odds of severe COVID-19 illness [41]. A study on obesity in veterans indicated that the state of residence was associated with higher odds of reporting overweight or obese status [28]. For our research study, geographic effects were mixed. Region 2 had higher odds of all morbidities other than mental health disorders versus Region 1, while Region 3 had universally lower odds than Region 1. Temporal effects were also mixed, and the effect sizes were small (odds ratios between 0.983 and 1.026).

As a start, future research should consider the effect of healthcare access, comorbidities, time in service, multiple deployments, jobs held in the service, and military rank on the health outcomes of veterans. For example, veterans who did not experience combat duty may have better health outcomes than veterans who had combat deployments. Veterans who served longer or had higher ranks may have different health statuses compared to veterans with fewer years of service and/or lower ranks. Multiple deployments may increase the odds of reporting the studied diseases. Separation from active duty could result in fewer routine medical office visits compared to the mandatory physical exams for active-duty personnel, which may impact veterans’ health outcomes. As previously indicated, several of the studied conditions are risk factors for other studied conditions, indicating more research is needed on the effect of comorbidities on veterans’ health status. Future research should also consider the job that the veteran performed while in the service; for example, have they been exposed to radiation (e.g., working on an aircraft carrier or in a submarine) or hazardous chemicals? Future research should analyze different data sets to determine if the low levels of self-reported mental health issues are unique to the studied BRFSS data set.

There are several practical implications derived from the study’s results. The US Department of Veterans Affairs should work to assess the effects of multiple comorbidities, multiple deployments, combat duty, and years in service on the health outcomes of veterans. The Veterans Affairs Department should evaluate the prevalence of service-related diseases reported in the study by the state to determine if more targeted healthcare resources are needed for the treatment of veterans in each state. They should develop interventions for veterans to assist them in managing and obtaining medical appointments because healthcare utilization for veterans is vital to obtaining positive health outcomes.

### Limitations

One limitation of this study is that the BRFSS data are self-reported, so they are not as accurate as measured data (i.e., clinical observations). Moreover, respondents may exhibit bias by over- or under-reporting on self-rating questions. Some respondents may not feel comfortable reporting their true depression levels because of social or job-related concerns. In the absence of the true values, whether the under- or over-reporting by the respondents is quite confirmed.

## 5. Conclusions

Overall, veterans continue to experience disproportionately higher odds of obesity/overweight status, heart disease, stroke, skin cancer, cancer, COPD, arthritis, kidney disease, and diabetes. Veterans do, however, report lower rates of mental health disorders. COVID-19 did not materially affect the higher odds that veterans experienced for all morbidities except mental health. These findings continue to suggest that there still exists a lack of effective interventions for the Veteran population from public agencies such as the US Veteran’s Administration (VA) and private organizations serving veterans (e.g., Veterans of Foreign Wars, American Legion).

Several implications are realized in the comparison of veteran versus non-veteran states in the periods before as well as after the COVID-19 pandemic. The prevalence of diseases (such as stroke, skin cancer, COPD, arthritis, kidney disease, or diabetes) is higher among veterans than among non-veterans. Such an imbalance is not rectified in the post-COVID-19 period. The mental health state among the veterans has been at parity with that of the non-veterans despite the COVID-19 pandemic.

## Figures and Tables

**Table 1 healthcare-11-02049-t001:** Sample and population estimates by veteran status 2003–2021.

Year	Non-Vet Population (Weighted)	Vet Population (Weighted)	%Veteran	Non-Vet Sample (Unweighted)	Vet Sample (Unweighted)	%Veteran
2003	190,348,049	30,003,072	13.62%	228,159	36,525	13.80%
2004	191,637,278	29,746,086	13.44%	260,982	42,840	14.10%
2005	194,578,583	29,532,523	13.18%	305,107	51,005	14.32%
2006	198,138,945	29,118,914	12.81%	304,989	50,721	14.26%
2007	202,498,717	27,673,461	12.02%	370,990	59,922	13.91%
2008	205,615,985	27,244,684	11.70%	358,433	56,076	13.53%
2009	208,756,506	26,249,349	11.17%	374,909	57,698	13.34%
2010	211,037,577	26,048,662	10.99%	390,643	60,432	13.40%
2011	212,198,501	25,812,791	10.85%	441,873	64,594	12.75%
2012	216,959,427	26,098,283	10.74%	415,817	59,870	12.59%
2013	219,968,409	26,056,006	10.59%	430,268	61,505	12.51%
2014	220,704,167	27,778,365	11.18%	402,544	62,120	13.37%
2015	224,174,518	27,172,620	10.81%	383,614	57,842	13.10%
2016	227,144,466	27,006,670	10.63%	422,384	63,919	13.14%
2017	229,254,924	26,398,281	10.33%	392,148	57,868	12.86%
2018	230,694,063	27,379,324	10.61%	381,382	56,054	12.81%
2019	226,740,688	25,689,603	10.18%	365,038	53,230	12.73%
2020	234,258,521	26,149,949	10.04%	353,737	48,221	12.00%
2021	222,097,968	23,943,672	9.73%	386,175	52,518	11.97%
Totals	3,610,450,803	465,008,694	11.41%	6,229,280	952,221	13.26%

**Table 2 healthcare-11-02049-t002:** Variables in the Study.

Variable	Description
**Dependent Variables**	
Overweight/Obese	Body mass index greater than 25.00? 1 = Yes, 0 = Otherwise
Angina or coronary heart disease	Had angina or coronary heart disease? 1 = Yes, 0 = Otherwise
Stroke	Had a stroke? 1 = Yes, 0 = Otherwise
Skin cancer	Had skin cancer? 1 = Yes, 0 = Otherwise
Other cancer	Had any other types of cancer? 1 = Yes, 0 = Otherwise
COPD	Had C.O.P.D., emphysema or chronic bronchitis? 1 = Yes, 0 = Otherwise
Arthritis	Arthritis, rheumatoid arthritis, gout, lupus, or fibromyalgia? 1 = Yes, 0 = Otherwise
Mental Health ^1^	1 = One or more days out of 30 of poor mental health, 0 = Otherwise
Kidney Disease ^2^	Had kidney disease? 1 = Yes, 0 = Otherwise
Diabetes	Had diabetes? 1 = Yes, 0 = Otherwise
**Demographic Controls**	
Age	Imputed age category: 1 = 18–24, 2 = 25–34, 3 = 35–44, 4 = 45–54, 5 = 55–64, 6 = 65+
Race	1 = Caucasian, 0 = Otherwise
Ethnicity	1 = Hispanic, 0 = Otherwise
Gender	1 = Birth Sex Male, 0 = Birth Sex Female
Marital Status	1 = Married, 0 = Otherwise
**Socioeconomic Controls**	
Income	Total household income: 1 = $75 K+, 0 = Otherwise
Education	1 = Graduated college or technical school, 0 = Otherwise
Employment	1 = Employed for wages, 0 = Otherwise
**Geographic Controls**	
Division	1 = New England, 2 = East North Central, 3 = East South Central, 4 = Middle Atlantic, 5 = Mountain, 6 = Pacific, 7 = South Atlantic, 8 = West North Central, 9 = West South Central, 10 = Territories
**Time**	Year, 2011–2021
**Independent Variables**	
Veteran Status	1 = Active, Reserve, or National Guard in United States Armed Forces, 0 = Otherwise
COVID-19 Period	0 = Prior to 2020, 1 = 2020, 2 = 2021

^1^ Includes depression, stress, emotions. ^2^ Not including kidney stones, bladder infections, or incontinence.

**Table 3 healthcare-11-02049-t003:** Age-adjusted percentages reported by health condition by year by Veteran status.

	Y2003	Y2004	Y2005	Y2006	Y2007	Y2008	Y2009	Y2010	Y2011	Y2012	Y2013	Y2014	Y2015	Y2016	Y2017	Y2018	Y2019	Y2020	Y2021	Y11-21
Overweight/Obese Non-Vet	0.56	0.57	0.58	0.58	0.59	0.60	0.60	0.61	0.60	0.60	0.60	0.59	0.59	0.59	0.59	0.60	0.60	0.59	0.59	0.60
Overweight/Obese Vet	0.70	0.70	0.70	0.71	0.72	0.73	0.73	0.73	0.73	0.72	0.72	0.72	0.71	0.71	0.71	0.72	0.71	0.70	0.71	0.71
Diabetes Non-Vet	0.11	0.12	0.13	0.13	0.14	0.14	0.14	0.14	0.15	0.15	0.16	0.16	0.16	0.17	0.17	0.17	0.17	0.17	0.17	0.16
Diabetes Vet	0.13	0.13	0.14	0.15	0.16	0.15	0.16	0.16	0.18	0.18	0.18	0.18	0.19	0.19	0.19	0.20	0.19	0.20	0.19	0.19
Heart Disease Non-Vet			0.08	0.08	0.08	0.08	0.07	0.08	0.08	0.08	0.08	0.08	0.07	0.08	0.07	0.07	0.07	0.07	0.07	0.07
Heart Disease Vet			0.13	0.14	0.13	0.14	0.13	0.14	0.13	0.13	0.13	0.13	0.12	0.12	0.12	0.12	0.11	0.12	0.11	0.12
Stroke Non-Vet			0.05	0.05	0.05	0.05	0.05	0.05	0.05	0.05	0.05	0.05	0.05	0.05	0.05	0.05	0.05	0.05	0.05	0.05
Stroke Vet			0.06	0.06	0.06	0.06	0.06	0.06	0.06	0.06	0.06	0.06	0.06	0.06	0.06	0.07	0.07	0.06	0.06	0.06
Skin Cancer Non-Vet									0.11	0.11	0.11	0.11	0.12	0.11	0.12	0.12	0.12	0.12	0.12	0.12
Skin Cancer Vet									0.16	0.16	0.16	0.17	0.17	0.16	0.17	0.17	0.18	0.16	0.18	0.17
Cancer Non-Vet									0.11	0.11	0.11	0.11	0.12	0.11	0.12	0.12	0.12	0.12	0.12	0.12
Cancer Vet									0.13	0.12	0.13	0.12	0.13	0.13	0.14	0.14	0.14	0.15	0.14	0.13
COPD Non-Vet									0.08	0.08	0.08	0.09	0.08	0.08	0.09	0.09	0.09	0.09	0.09	0.09
COPD Vet									0.09	0.09	0.10	0.10	0.10	0.10	0.11	0.11	0.11	0.12	0.11	0.10
Kidney Disease Non-Vet									0.03	0.04	0.04	0.04	0.04	0.04	0.04	0.05	0.05	0.05	0.05	0.04
Kidney Disease Vet									0.03	0.04	0.04	0.04	0.04	0.05	0.05	0.05	0.05	0.06	0.05	0.05
Arthritis Non-Vet									0.37	0.38	0.38	0.39	0.37	0.38	0.37	0.38	0.37	0.37	0.37	0.37
Arthritis Vet									0.36	0.38	0.37	0.37	0.37	0.38	0.37	0.39	0.37	0.38	0.39	0.37
Mental Health Non-Vet	0.37	0.37	0.36	0.37	0.36	0.36	0.36	0.36	0.37	0.37	0.35	0.35	0.36	0.36	0.38	0.39	0.41	0.40	0.44	0.38
Mental Health Vet	0.31	0.31	0.30	0.30	0.28	0.30	0.30	0.30	0.30	0.30	0.29	0.29	0.30	0.30	0.31	0.32	0.35	0.33	0.38	0.31

Color legend: Veterans > Non-Veterans (depicted in RED), Veterans < Non-Veterans (depicted in GREEN).

**Table 4 healthcare-11-02049-t004:** Results of the Quasibinomial General Linear Models (Odds-Ratios).

Variable	Overweight/Obese		Heart Disease		Stroke		Skin Cancer		Cancer		COPD		Arthritis		Mental Health		Kidney Disease		Diabetes	
Age 25–34	1.96	***	2.02	***	2.57	***	1.47	***	2.60	***	1.88	***	2.89	***	0.88	***	1.78	***	2.34	***
Age 35–44	2.71	***	4.44	***	5.42	***	3.59	***	4.31	***	2.95	***	6.30	***	0.78	***	2.83	***	6.72	***
Age 45–54	3.16	***	11.20	***	10.39	***	8.61	***	7.78	***	4.92	***	13.04	***	0.67	***	4.34	***	15.09	***
Age 55–64	3.41	***	20.83	***	14.99	***	15.92	***	12.86	***	6.69	***	22.03	***	0.52	***	6.01	***	24.80	***
Age 65+	2.99	***	31.34	***	18.68	***	33.68	***	22.15	***	5.92	***	27.96	***	0.26	***	7.81	***	30.10	***
Caucasian	0.97	***	1.15	***	0.76	***	6.46	***	1.37	***	1.31	***	1.21	***	1.26	***	0.86	***	0.59	***
Hispanic	1.17	***	0.90	***	0.66	***	1.25	***	0.84	***	0.67	***	0.70	***	0.87	***	1.03	^NS^	1.03	*
Male	1.81	***	1.68	***	1.09	***	1.04	***	0.63	***	0.82	***	0.65	***	0.62	***	0.96	*	1.21	***
Married	1.06	***	0.86	***	0.71	***	1.09	***	0.95	***	0.64	***	0.85	***	0.67	***	0.83	***	0.94	***
Income > $75K	1.10	***	0.76	***	0.59	***	1.18	***	0.95	***	0.52	***	0.81	***	0.88	***	0.75	***	0.70	***
College Grad	0.68	***	0.74	***	0.64	***	1.25	***	1.01	*	0.47	***	0.68	***	0.98	**	0.79	***	0.65	***
Wage Employee	1.19	***	0.47	***	0.35	***	0.76	***	0.65	***	0.47	***	0.62	***	0.77	***	0.47	***	0.67	***
Region 2	1.10	***	1.23	***	1.28	***	1.45	***	1.03	*	1.30	***	1.17	***	0.91	***	1.04	*	1.22	***
Region 3	0.80	***	1.00	^NS^	0.87	***	0.96	*	0.97	^+^	0.88	***	0.89	***	0.92	***	0.85	***	0.92	***
Region 4	0.79	***	0.81	***	0.87	***	1.46	***	0.97	*	0.84	***	0.85	***	1.00	^NS^	1.03	^+^	0.83	***
Region 5	0.77	***	0.89	***	0.82	***	1.06	***	1.02	*	0.86	***	0.89	***	0.97	***	0.86	***	0.87	***
Region 6	0.75	***	0.84	***	0.85	***	1.40	***	0.99	^NS^	0.78	***	0.79	***	1.06	***	0.95	*	0.86	***
Region 7	0.88	***	1.03	*	1.02	^+^	1.53	***	0.98	^NS^	1.02	^+^	0.91	***	0.87	***	1.00	^NS^	0.99	^NS^
Region 8	1.02	^+^	1.92	***	0.58	***	0.85	**	0.94	^+^	0.82	***	1.01	^NS^	0.55	***	0.77	***	0.96	*
Region 9	0.97	***	0.92	***	0.96	*	1.06	***	0.98	^NS^	0.85	***	0.85	***	0.84	***	0.87	***	0.93	***
Region 10	0.97	**	1.05	**	1.09	***	1.27	***	0.97	^+^	0.97	^+^	0.88	***	0.88	***	1.04	^+^	1.07	***
Year	0.99	**	0.98	***	1.02	***	1.00	*	1.00	**	1.01	***	0.99	**	1.01	***	1.03	***	1.01	***
Veteran	1.26	***	1.33	***	1.27	***	1.30	***	1.48	***	1.33	***	1.25	***	0.99	^NS^	1.12	***	1.11	***
COVID	1.00	*	0.99	^NS^	0.97	**	0.98	*	0.99	^+^	0.97	***	0.98	**	1.06	***	0.95	***	0.98	*
Veteran × COVID	0.98	^+^	0.98	^NS^	0.97	^NS^	1.01	^+^	1.01	^NS^	1.05	***	1.03	***	0.99	^NS^	1.04	**	0.99	^NS^

*** *p* < 0.001, ** *p* < 0.01, * *p* < 0.05, ^+^ *p* < 0.10, ^NS^ = not significant.

## Data Availability

Data are made publicly available by the CDC on the BRFSS website. https://www.cdc.gov/brfss/annual_data/annual_data.htm.

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
