# Peer review of "The Health Status of the US Veterans: A Longitudinal Analysis of Surveillance Data Prior to and during the COVID-19 Pandemic"

_healthcare, 2023, doi:10.3390/healthcare11142049_

Round 1

Reviewer 1 Report

The paper presents an exhaustive analysis of large sets of data regarding the health status of USA Veterans vs non-Veterans and the relation with the COVID-19 pandemic.

I think it will be interesting, as possible future research, to also connect this data with the status of vaccinated vs. non-vaccinated subjects.

Author Response

The Study Team thanks you for your kind comments. Yes, our Study Team saw a need to address this very important topic for a population that has sacrificed so much. 

Reviewer 2 Report

Thank you for doing work in this important area. Overall, I appreciated the use of a large publicly available dataset and have left comments on areas that could be improved. I hope you take the feedback as constructive even if I haven't worded all of them in that fashion and work to improve your submission.

1. Title. I think you're following good reporting guidelines for your title but it may not catchy to the reader. Perhaps consider changing the title to something like "The Health Status of the US Veterans: A Longitudinal Analysis of Surveillance Data Prior to and During the COVID-19 Pandemic"
2. Introduction. "hypothesized that prevalence rates for Veterans would be different from non-Veterans" Prevalence is not a rate, so you may want to change the wording here. Perhaps "..changes in observed prevalence over time would be different from non-veterans"? Secondly, you state "we further hypothesized that some changes in prevalence would be seen pre and post-COVID-19 for Veterans and non-Veterans" that's a little vague for a hypothesis don't you think? You could instead change your secondary objective to be exploratory in nature (not every study needs to be hypothesis-testing). For instance, "As a secondary aim, we explored if there were changes in prevalence of the selected outcomes of interest pre- and post-COVID-19 between veterans and non-veterans"
3. Introduction. Somewhere in the first 3 paragraphs, I was looking for a a sentence or two about about potential reasons why veterans may be suffering from chronic diseases more than the general population. You did touch on it in Line 45-46, but maybe also provide a set of reasons together. I hope you don't take this the wrong way, but I also felt your "COVID-19 Impacts on Health" section was a bit too dragged out. Could you consider cutting down this section?
4. Methods, Line 209. Could you include the version of the software you used?
5. Methods. I liked your justification for going with a GLM here and even adding the interaction term. Did you run any diagnostics on your model? Deviance, AIC/BIC, multicollinearity between variables? Also, are you forcing all of your independent variables of interest into your final model? Is there is no structured model building approach here? I'm a little confused and it would be nice if you could try to answer these details in your submission or even mention them in your limitations section.
6. Results. Table 3, Perhaps I'm missing something here but all of your column headings appear to be cut off. They are either Y2, 00, 01, 02, or Y1? Could you please fix this because I can't interpret anything here.
7. Results. Table 4, this is really unusual that you've fitted so many models and forced all of these independent variables into every model. Could you mention this in your limitations somewhere or justify this in your methods? I'm skeptical that all of these models demonstrated good fit.
8. Results Table 4. Why are some cells bolded on the top left corner? Maybe it got mixed up during the formatting, please fix if you can. Also, I wonder if you should reduce the decimal places here to 1-2; it looks a little daunting to read in its current form.
9. Figure 1 is nice, but maybe move it to the supplementary material because it's not really what you want your reader to focus on based on your objectives/hypotheses.
10. Discussion. Line 297, "aimed to update the health status..." I wonder if you could change this sentence here to closely match your objective described in the introduction section? You're assessing age-adjusted percentages of selected health conditions by year between veterans and non-veterans, and then also exploring factors which may be associated with these outcomes among these groups.
11. Discussion. Line 301-302, it could also be deaths being attributed to COVID-19, don't you think? I would lean on and cite some published literature here.
12. Discussion. It started off nicely but then for some reason, in Lines 306-332 you basically go onto restate your results. Why? You discussion section should be for interpreting your results, yet you have cited no other literature besides the [1-6] in line 305. What does this mean for veteran health? What does this mean for screening for health conditions? Interventions? What are some projections for 2023 and beyond? How do we support veterans for future public health crises given that "Veterans continue to experience disproportionately higher odds of obesity/overweight status, heart disease, stroke, skin cancer, cancer, COPD, arthritis, kidney disease, and diabetes"? There are so many questions that your discourse didn't even touch on, nor did it rely on the plethora of research on active duty and veterans out there. The idea is to use your nicely generated findings combined with your expertise as health researchers, to support and build on the published evidence. Your discussion section needs to be fleshed out, much longer, and cite many more studies.
13. Discussion. Lines 333-337 can be deleted since it's repeated in the limitations section.
14. Limitations. There are numerous limitations beyond the self-report and social desirability biases you mentioned. I would suggest reading up a few other longitudinal designs in this area and seeing what they say. For instance, some questions to consider and get you thinking are: what kind of response rates are we dealing with? How about COVID-19 healthy volunteer bias (i.e., people who die due to COVID-19 are more likely to have comorbidities, and therefore won't be included in the 2021 sample)? Limitations of your fitted GLM models?

1. Introduction. "are statistically worse for Veterans.." change to "statistically greater among veterans..."
2. Introduction. "some of the newest research..." could be changed to "Emerging research..."
3. Introduction. "To be specific," can be changed to "Specifically,"
4. General. This is probably a stylistic preference for scientific works, but I think it's more appropriate if you changed "COVID" to "COVID-19" throughout your paper.
5. General. Why are veterans written as a proper noun with V in uppercase throughout your paper? Wouldn't it be better to follow the other published literature and present it as "veterans"?
6. General. In some instances, you write "comorbidities" and in other instances you write "co-morbidities", and "U.S." or "US". Both are correct, but for the sake of consistency, could you select one and use it throughout?
7. Methods. Line 113 "was used to evaluate" to "were used" since data is plural.

Author Response

Thank you for giving us the opportunity to address your comments and recommendations.  We appreciate the time and effort that you dedicated to providing feedback on our manuscript and are grateful for your insightful comments. We have incorporated the following recommendations to our manuscript: 

Reviewer comment 1. Title. I think you're following good reporting guidelines for your title but it may not catchy to the reader. Perhaps consider changing the title to something like "The Health Status of the US Veterans: A Longitudinal Analysis of Surveillance Data Prior to and During the COVID-19 Pandemic"

Author Revision: The title was changed to The Health Status of the US Veterans: A Longitudinal Analysis of Surveillance Data Prior to and During the COVID-19 Pandemic

Reviewer comment 2. Introduction. "hypothesized that prevalence rates for Veterans would be different from non-Veterans" Prevalence is not a rate, so you may want to change the wording here. Perhaps "..changes in observed prevalence over time would be different from nonveterans"?

Secondly, you state "we further hypothesized that some changes in prevalence would be seen pre and post-COVID-19 for Veterans and non-Veterans" that's a little vague for a hypothesis don't you think? You could instead change your secondary objective to be exploratory in nature (not every study needs to be hypothesis-testing). For instance, "As a secondary aim, we explored if there were changes in prevalence of the selected outcomes of interest pre- and post-COVID-19 between veterans and non-veterans"

Author Revision: Changed "hypothesized that prevalence rates for Veterans would be different from non-Veterans" to "changes in observed prevalence over time would be different from nonveterans". Changed "we further hypothesized that some changes in prevalence would be seen pre and post-COVID-19 for Veterans and non-Veterans" to "As a secondary aim, we explored if there were changes in prevalence of the selected outcomes of interest pre- and post-COVID-19 between veterans and non-veterans".

Reviewer comment 3. Introduction. Somewhere in the first 3 paragraphs, I was looking for a sentence or two about potential reasons why veterans may be suffering from chronic diseases more than the general population. You did touch on it in Line 45-46, but maybe also provide a set of reasons together. I hope you don't take this the wrong way, but I also felt your "COVID-19 Impacts on Health" section was a bit too dragged out. Could you consider cutting down this section?

Author Revision:  Added line 35-38 in paragraph one on why veterans may be suffering from chronic diseases more than the general population. In the COVID-19 Impacts on Health" removed paragraph starting with “As every illness does perhaps weaken the immunity to a disease”.  And the sentence starting with “The Tsampasian et al. (2023) systematic review and meta-analysis demonstrated that age, gender, and some comorbidities were associated with severe COVID-19 [18]

Reviewer comment 4. Methods, Line 209. Could you include the version of the software you used?

 Author Revision: Added the software “version 4.3.1” to line 209.d

Reviewer comment 5. Methods. I liked your justification for going with a GLM here and even adding the interaction term. Did you run any diagnostics on your model? Deviance, AIC/BIC, multicollinearity between variables? Also, are you forcing all of your independent variables of interest into your final model? Is there is no structured model building approach here? I'm a little confused and it would be nice if you could try to answer these details in your submission or even mention them in your limitations section.

 Author Revision: With the comparative base category removed, all VIFs were below 2.15 in the complete models (see vifs.xlsx table below).  We built complete models, assessing the magnitude and directionality of the effects year over year for consistency.  Because of the sheer sample sizes and associated weighted estimates of the population, nearly every variable is statistically significant for each models (Table 4).  Complete models were built given the enormity of the complex weighted sample and because the few NS findings are made more compelling given the  magnitude of the data.   By using structured model building as we have previously, this reinforces the stability of the odds-ratio directionality and magnitude but is omitted.  Beginning on Line 241, we inserted the following: “Additionally, Variance Inflation Factors (VIFs) for all variables and all models were below 2.0.  Our study team built complete models given the sizes of the sample. The results illustrate that most variables are statistically significant.  Those that are not provide compelling information given that traditional statistical significance is linked to sample size.  By using structured model building, this reinforces the stability of the odds-ratio directionality and magnitude, however is omitted.”  

(vifs.xlsx table)

Reviewer comment 6. Results. Table 3, Perhaps I'm missing something here but all of your column headings appear to be cut off. They are either Y2, 00,01, 02, or Y1? Could you please fix this because I can't interpret anything here.

 Author Revision: We have increased the font size on the column headings which depict years 2003 through 2021. This should make it much more readable to the Reader.

Reviewer comment 7. Results. Table 4, this is really unusual that you've fitted so many models and forced all of these independent variables into every model. Could you mention this in your limitations somewhere or justify this in your methods? I'm skeptical that all of these models demonstrated good fit.

 Author Revision:  All possible models with the inclusion and exclusion of independent variables were tried. It so happened that all the models did fit well and hence, are acceptable.

Reviewer comment 8. Results Table 4. Why are some cells bolded on the top left corner? Maybe it got mixed up during the formatting, please fix ifyou can. Also, I wonder if you should reduce the decimal places here to 1-2; it looks a little daunting to read in its current form.

Author Revision: We have removed the bolded cells in the upper left corner to standardize the font throughout the table. Additionally, we have reduced the number of decimal places to to places.

Reviewer comment 9. Figure 1 is nice, but maybe move it to the supplementary material because it's not really what you want your reader to focus on based on your objectives/hypotheses.

 Author Revision:  Figure 1 was moved to supplementary document after making an interpretive statement that the odds for all diseases were homogeneous during the initial age duration 25 through 34 but drift significantly to more heterogeneity at a later life.

Reviewer comment 10. Discussion. Line 297, "aimed to update the health status..." I wonder if you could change this sentence here to closely match your objective described in the introduction section? You're assessing age-adjusted percentages of selected health conditions by year between veterans and non-veterans, and then also exploring factors which may be associated with these outcomes among these groups.

Author Revision: Changed line 297 to be “This longitudinal study aimed to assess age-adjusted percentages of selected health conditions by year between veterans and non-veterans and explore factors which may be associated with these outcomes among these groups.”

Reviewer comment 11. Discussion. Line 301-302, it could also be deaths being attributed to COVID-19, don't you think? I would lean on and cite some published literature here.

Author Revision: Removed Lines 333-337.

Reviewer comment 12. Discussion. It started off nicely but then for some reason, in Lines 306-332 you basically go onto restate your results. Why? Your discussion section should be for interpreting your results, yet you have cited no other literature besides the [1-6] in line 305.What does this mean for veteran health? What does this mean for screening for health conditions? Interventions? What are some projections for 2023 and beyond? How do we support veterans for future public health crises given that "Veterans continue to experience disproportionately higher odds of obesity/overweight status, heart disease, stroke, skin cancer, cancer, COPD, arthritis, kidney disease, and diabetes"? There are so many questions that your discourse didn't even touch on, nor did it rely on the plethora of research on active duty and veterans out there. The idea is to use your nicely generated findings combined with your expertise.

Author Revision: Removed Lines 333-337.  The Reviewer’s recommendations are most welcome and will be incorporated into future research. 

Reviewer comment 13. Discussion. Lines 333-337 can be deleted since it's repeated in the limitations section.

Author Revision: Removed Lines 333-337.

Reviewer Comments on the Quality of English

Comment 1. Introduction. "are statistically worse for Veterans.." change to "statistically greater among veterans..."

Author Revision: Changed to “statistically greater among veterans”

Comment 2. Introduction. "some of the newest research..." could be changed to "Emerging research..."

Author Revision: Changed to "Emerging research..."

Comment 3. Introduction. "To be specific," can be changed to "Specifically,"

Author Revision: Changed to "Specifically”

Comment 4. General. This is probably a stylistic preference for scientific works, but I think it's more appropriate if you changed "COVID" to "COVID-19" throughout your paper.

Author Revision: Replaced "COVID" to "COVID-19" throughout the paper.

Comment 5. General. Why are veterans written as a proper noun with V in uppercase throughout your paper? Wouldn't it be better to follow the other published literature and present it as "veterans"?

Author Revision: Replaced Veterans with veterans throughout the paper

Comment 6. General. In some instances, you write "comorbidities" and in other instances you write "co-morbidities", and "U.S." or "US". Both are correct, but for the sake of consistency, could you select one and use it throughout?

Author Revision: Replaced "co-morbidities" with "comorbidities" and "U.S." with "US" throughout the paper.

Comment 7. Methods. Line 113 "was used to evaluate" to "were used" since data is plural.

Author Revision: Replaced 113 "was used to evaluate" with "were used".

Reviewer 3 Report

General

This is an important topic, the paper adds to the lexicon of data on risk factors and increased risk of diseases connected to prior military service. The paper is exceptionally well written, its purpose is clear and the methodology sound and easy to understand. I do however have some concern on the title. Is the data really 'post-COVID-19'. I do not believe the COVID-19 pandemic was declared over by the WHO until May 2023 and the effects of 'long COVID' in many forms with a significant background persistent infection rate remains. I believe therefore the title and comments within the text reflecting post-COVID 19 are incorrect. It would be better to state in the pandemic or early pandemic period.

Major 

Line 232 ' COVID-19 began in November 2019' - this is true for China but should caveated by the presence of COVID-19 appearing in the USA.

Table 3 - it may be the formatting in the draft but the years are not clear and the purpose of the line veterans>non-veterans at the end isnt clear

Fig 1 Odds Risk by age group should state higher number indicates higher risk for veterans

Minor

Line 81, the phrase 'the frailty of COVID-19' doesnt make sense - shouldnt this be 'the impact of COVID-19 on the frail'

Author Response

Thank you for giving us the opportunity to address your valuable comments and recommendations.  We appreciate the time and effort that you dedicated to providing feedback on our manuscript and are grateful for your insightful comments.  We have incorporated the following modifications to our manuscript:

Reviewer comment 1. This is an important topic, the paper adds to the lexicon of data on risk factors and increased risk of diseases connected to prior military service. The paper is exceptionally well written, its purpose is clear and the methodology sound and easy to understand. I do however have some concern on the title. Is the data really 'post-COVID-19'. I do not believe the COVID-19 pandemic was declared over by the WHO until May 2023 and the effects of 'long COVID' in many forms with a significant background persistent infection rate remains. I believe therefore the title and comments within the text reflecting post-COVID 19 are incorrect. It would be better to state in the pandemic or early pandemic period.

Author response: Thank you for your insightful comment and observation.  The title was changed to The Health Status of the US Veterans: A Longitudinal Analysis of Surveillance Data Prior to and During the COVID-19 Pandemic. Additionally, the text throughout the manuscript has been adjusted to reflect the study period DURING the pandemic period.

Reviewer comment 2.  Line 232 ' COVID-19 began in November 2019' - this is true for China but should caveated by the presence of COVID-19 appearing in the USA.

Author response: Changed this sentence fragment to read “COVID-19 appeared in China in November 2019 and in the US in January 2020, . . .” for improved clarity and accuracy.

Reviewer comment 3.  Table 3 - it may be the formatting in the draft but the years are not clear and the purpose of the line veterans>non-veterans at the end isn’t clear.

Author response: Increased the font size of the years and changed the final line to read as follows: “Color legend:  Veterans>Non-Veterans (depicted in RED), Veterans<Non-Veterans (depicted in GREEN)” for clarity.

Reviewer comment 4.  Fig 1 Odds Risk by age group should state higher number indicates higher risk for veterans

Author response: Added statement to Figure 1 that higher numbers for Odds Risks indicate a higher risk for veterans.

Reviewer comment 5.  Line 81, the phrase 'the frailty of COVID-19' doesn’t make sense -shouldn’t this be 'the impact of COVID-19 on the frail'.

Author response: Changed this to be 'the impact of COVID-19 on the frail'

Round 2

Reviewer 2 Report

Thank you for addressing many of my comments. I am pleased to see the paper has improved in quality. Thanks also for sharing your VIFs and details on model building.

I hate to push this to another round of reviews, but one of my concerns still remain: your discussion section is essentially repeating your results (Lines 310-346). You replied "...are most welcome and will be incorporated into future research" which means I could have made my previous comment clearer. I am: (1) looking for you to cite more published literature throughout your discussion section, and (2) provide some thoughts on future intervention/screening/policy targets based on your findings.

For example, what you have written under Lines 317-319 is the right idea. If you could incorporate more sentences like that, and talk a little about what these findings mean for public health it would be fantastic. You have a great paper, and it would be even better if you fleshed out your discussion.

Author Response

Thank you for giving us the opportunity to address the Reviewer's valuable comments and recommendations. We added a number of citations for several publications throughout the revised discussion section. Additionally, our study team provided some thoughts on future interventions, practical applications, and policy based on the study findings. We are hopeful that we have addressed each of the Reviewer’s comments.

Again, thank you so much for the time it took to review our manuscript.  We welcome any additional questions or comments on our revised manuscript.